# Efficacy of Artificial Intelligence in the Categorisation of Paediatric Pneumonia on Chest Radiographs: A Systematic Review

**DOI:** 10.3390/children10030576

**Published:** 2023-03-17

**Authors:** Erica Louise Field, Winnie Tam, Niamh Moore, Mark McEntee

**Affiliations:** 1Discipline of Medical Imaging and Radiation Therapy, University College Cork, College Road, T12 K8AF Cork, Ireland; 2Department of Midwifery and Radiography, University of London, Northampton Square, London EC1V 0HB, UK

**Keywords:** artificial intelligence (AI), deep learning (DL), paediatric pneumonia, chest radiograph, computer-aided detection (CAD)

## Abstract

This study aimed to systematically review the literature to synthesise and summarise the evidence surrounding the efficacy of artificial intelligence (AI) in classifying paediatric pneumonia on chest radiographs (CXRs). Following the initial search of studies that matched the pre-set criteria, their data were extracted using a data extraction tool, and the included studies were assessed via critical appraisal tools and risk of bias. Results were accumulated, and outcome measures analysed included sensitivity, specificity, accuracy, and area under the curve (AUC). Five studies met the inclusion criteria. The highest sensitivity was by an ensemble AI algorithm (96.3%). DenseNet201 obtained the highest level of specificity and accuracy (94%, 95%). The most outstanding AUC value was achieved by the VGG16 algorithm (96.2%). Some of the AI models achieved close to 100% diagnostic accuracy. To assess the efficacy of AI in a clinical setting, these AI models should be compared to that of radiologists. The included and evaluated AI algorithms showed promising results. These algorithms can potentially ease and speed up diagnosis once the studies are replicated and their performances are assessed in clinical settings, potentially saving millions of lives.

## 1. Introduction

Pneumonia is one of the leading causes of global mortality and morbidity [1] and the leading cause of death among children under five years of age [2]. The chest X-ray (CXR) is the primary diagnostic tool in both the detection and diagnosis of paediatric pneumonia [1] due to the unspecific and subjective signs and symptoms of the infection [3], while sputum cultures are often extremely difficult to ascertain [4]. A lack of expert radiologists, particularly in resource-constrained countries, where paediatric pneumonia is endemic with shockingly high mortality rates [5], might have significantly contributed to the high mortality rates.

Pneumonia, as a whole, often manifests on a chest radiograph as areas of increased opacity [6]. Bacterial and viral pneumonia, the two most common aetiologies, have different appearances on CXRs and have different treatment regimes [7]. Bacterial pneumonia often manifests as a lobar and focal consolidation, whereas viral pneumonia can present as an interstitial pattern. Although the different appearances on CXRs, interpretation of the aetiologies on CXR varies amongst physicians [3] as the opacification presented is often variable and irregular [8].

There are roughly two billion CXRs performed in the United States annually [9], with approximately two million of these being on paediatric patients [10]. The accumulation of imaging data and the increasing complexity of medical history pose new challenges in modern medicine but also open up new opportunities in implementing artificial intelligence (AI) for effective detection and diagnosis [3].

AI is defined, by the father of AI, Marvin Minsky, as “the science of making machines do things that would require intelligence if done by men” [11]. Machine learning (ML) and deep learning (DL) are both under the umbrella of AI. In ML, the system identifies patterns by learning from data and makes decisions with minimal programming and human interventions [12,13]. On the other hand, DL involves multiple processing layers, with individual layers extracting a large number of features from unstructured data before progressing to the next layer [14]. Higher levels typically represent more abstract concepts [15] and can give a more comprehensive depiction or decision after passing through the entire network [14].

Computer-aided detection (CAD) was first introduced between 1963–1973 [16,17,18,19,20,21], and utilised in facets of radiology, predominantly the identification of lung, colorectal, breast, and prostate cancer, over the past 20 years [22,23,24,25,26,27,28,29,30,31,32,33]. CAD is trained in a regimental fashion and can only be improved by inputting more data, while AI has the autonomous learning element in which explicit programming and human instructions are not necessary for its improvement [34].

AI interpretation of medical images can also potentially be utilised in low- to middle-income countries, where the availability of radiologists is limited. Currently, interpretations rely on teleradiology [35], though it is not flawless. Cross-border teleradiology, in particular, is challenging due to liability in case of malpractice, healthcare professional registration restrictions, data protection, quality of the reporting, healthcare system, and cultural differences [36]. This is where AI can potentially come in to make the diagnostic and treatment pathway of patients smoother, especially in these developing nations [35].

Recent studies exhibited that AI algorithms outperformed radiologists in the detection of skin cancer [37], diabetic retinopathy [38], and haemorrhage identification [39], probably owing to the recent AI model advancement [40] and widening availability of electronic health record, providing more training materials. The winner of the Turning Prize said in 2016, “We should stop training radiologists now. It’s just completely obvious that within five years, DL is going to do better than radiologists” [41].

These successes have sparked interest in the automated diagnosis of paediatric pneumonia in CXRs, reflected by the increased number of studies regarding the diagnostic accuracy of AI for paediatric pneumonia over the last number of years. When being asked about AI’s role in diagnosing pneumonia, Andrew Ng, the co-founder and head of Google Brain, went further—“radiologists should be worried about their jobs” [42]. A recent study by Kermany et al. [43] showed that a customised AI model demonstrated a good level of classification of paediatric pneumonia on CXR. By systematically reviewing the current literature regarding AI and paediatric pneumonia, the current gap in literature may be filled and potentially result in a dramatic improvement of accuracy and efficiency in differentiating bacterial and viral pneumonia in paediatric CXRs in a clinical setting [44].

## 2. Materials and Methods

### 2.1. Literature Search Strategy

A comprehensive search of the literature was conducted in accordance with the Preferred Reporting Items for Systematic Reviews and Meta-Analyses (PRISMA) guidelines [45] in February 2021. A number of pre-determined keywords were pooled for this systematic search, and subsequent medical subject headings (MeSH terms) were generated. The MeSH terms used for this systematic search include: (‘artificial intelligence’ OR ‘deep learning’ OR ‘CNN’ OR ‘convolutional neural network’ OR ‘deep residual network’), (‘paediatric pneumonia’ OR ‘pediatric pneumonia’ OR ‘child* pneumonia’), ‘classification’, (‘chest xray’ OR ‘chest x ray’ OR ‘chest x-ray’ OR ‘CXR’ OR ‘chest radiograph’). The pre-determined MeSH terms were then linked using Boolean operators specific to each database (PubMed, Science Direct, Embase, ProQuest, and Scopus) to retrieve all relevant articles evaluating the diagnostic efficacy of AI models in the classification of paediatric pneumonia in CXRs. These databases were chosen as opposed to others due to their scientific reliability and coverage. Google Scholar and the reference lists of relevant studies were also screened to ensure as much grey literature was captured as possible. In addition to these criteria, only peer-reviewed articles published in English were included in this review.

### 2.2. Study Selection and Eligibility Criteria

The initial selection involved screening the paper title and abstract. After gathering all of the potential papers, a full-text assessment was undertaken. Studies were included if they met the following inclusion criteria: only original cross-sectional studies, cohort or case-control studies, randomised control trials (RCTs), or diagnostic accuracy studies in the format of either journal articles, dissertations, conference proceedings, or grey literature, disseminated between 2015–2021, were included. In addition, the paper must analyse/evaluate the AI performances on the classification of pneumonia aetiology (bacterial or viral) using CXR test datasets of children under the age of 16 years in acute healthcare settings. These studies should also evaluate the AI performance using at least three of the following parameters: accuracy, sensitivity, specificity, and area under the curve (AUC). Studies which did not meet one of the inclusion criteria, or inaccessible full-text articles, were excluded.

### 2.3. Data Extraction

Data extraction was performed on included studies using the amended version of the Cochrane ‘Data collection form for Intervention Reviews for RCTs and Non-RCTS—Template’ [46]. A general overview of the data extracted is as follows: general information (report title, study ID, date form completed, etc.), study eligibility, characteristics of study (the aim of the study, design, etc.), participants (description of images from dataset, setting, age, etc.), AI model (AI type, mode, detail, etc.), outcomes (sensitivity, specificity, accuracy, AUC), and other information (key conclusions, future work, etc.). Each data extraction form was reviewed by two independent reviewers, and any discrepancies were resolved by consensus. The original data extraction form applied to each included study can be obtained from the author on request.

### 2.4. Quality Assessment and Risk of Bias

A quality assessment of each included study was examined using the Critical Appraisal Skills Programme (CASP) Diagnostic Study Checklist, which focuses on the result validity, measuring parameters and transparency, and generalisability [47]. A risk of bias (ROB) assessment was also completed for each included study. The tool utilised for the ROB was adjusted from Hung et al.’s QUADAS-2 tool, whose systematic review investigated the use and performance of AI applications in the maxillofacial and dental radiology [48]. This adapted ROB consists of four key domains: patient selection, index test, reference standard, and study flow and timing with regard to both applicability and general ROB. Each domain was assessed using a three-point scale, low (green), high (red), or unclear (yellow), to reflect the level of bias concerns accordingly. All CASP and ROB checklists were reviewed by two independent reviewers, and any discrepancies were resolved by consensus.

## 3. Results

### 3.1. Study Selection

A total of 114 papers were identified following the initial systematic search from the six databases previously mentioned, combined with subsequent manual searches of reference lists based on the relevance of their title to the research question. After the removal of duplicates, 82 papers were considered for abstract screening, and 24 out of 82 were considered suitable and underwent full-text assessment. Of these 24 papers, 19 did not meet the required inclusion criteria due to a broad spectrum of reasons, such as the study did not classify pneumonia subtypes, the study did not include the desired (number of) outcome measures, and the dataset used was inapplicable. Five papers were included at last. The PRISMA flowchart exhibiting the study eligibility and selection process is presented in Figure 1. The two independent reviewers agreed with the initial search and study selection/eligibility process, with no discrepancies found.

### 3.2. Study Quality Assessment

The quality assessment of each included study can be seen in Figure 2. All five papers were considered as having a “low” risk of concern in the workflow, and both reference standard domains. Gu et al.’s study [44] was regarded as a “high” risk in the subject selection domain because the authors omitted CXRs without the condition of interest. In the index test domain, Rajaraman et al.’s [5] and Karthikeyan’s studies [49] were graded as “unclear”, while Ferreira et al.’s study [50] was considered a “high” risk, in the same domain due to the fact that the performance of the AI model was not evaluated by an independent testing dataset that was excluded in the development of the AI model. Since the QUADAS-2 tool allows a study to contain one element ascertaining a high risk of bias without being eliminated [48], none of the selected studies were eliminated at this stage.

### 3.3. Study Characteristics

Table 1 presents the characteristics of the included papers. All included papers utilised the same public dataset obtained from Guangzhou Women and Children’s Medical Centre; however, each paper varied regarding the number of images used for training, testing and validation sets. All studies, apart from Gu et al.’s study [44], included ‘normal’ radiographs, as well as bacterial and viral pneumonia, labelled radiographs in their dataset. There were some similarities and overlaps with regard to pre-processing methods employed, though none of the studies utilised precisely the same pre-processing strategy.

### 3.4. Diagnostic Accuracy of AI Algorithms in Distinguishing Viral Pneumonia from Bacterial Pneumonia

Table 2 presents the diagnostic accuracy measures (sensitivity, specificity, accuracy, and AUC) of each AI algorithm employed in respective papers. The ensemble set created achieved the highest sensitivity value of 96.3% [50] across all artificial algorithms examined (95% confidence intervals is not stated in the article). The DenseNet201 algorithm described by Karthikeyan reigned supreme with regard to both specificity and accuracy value obtaining 94% and 95%, respectively [49]. The AUC value closest to 1 was achieved by the VGG16 architecture (AUC = 0.9536) [5]. The baseline image, as well as the cropped ROI, scored 0.962 [5].

## 4. Discussion

CXR is routinely performed around the world to diagnose both subsets of pneumonia in paediatric patients [5]. Due to the complexity of lung diseases, the diagnosis of pneumonia on chest radiographs heavily relies on the eyes of a veteran radiologist. Therefore, there is a huge potential for AI algorithms to assist and further improve detection. This study aims to combine the results of all published literature focused on the classification of sub-types of paediatric pneumonia on chest X-rays using deep learning algorithms.

In this study, the efficacy of respective AI models in the classification of paediatric pneumonia on chest radiographs was evaluated by assessing accuracy, sensitivity, specificity, and AUC. Regarding diagnostic accuracy and specificity, the deep learning algorithm, the DenseNet201 model utilised by Karthikeyan’s study, performed far superiorly to the rest, yielding results of 95% and 94%, respectively [49]. AUCs measure the ability of a test (the individual algorithm in this case) to distinguish the presence or absence of a specific pathology [52] and take sensitivity and specificity into consideration. Thus, AUC can indicate how well a classifier is performing. The AI model, the customised VGG16 employed by Rajaraman et al. [5], achieved 96.2% in AUC for both of the baseline image sets (i.e., the original chest radiographs produced and cropped images), raising the question as to whether or not images that are cropped (just include the ROI) aid/improve AI models in the classification of paediatric pneumonia, or if it is simply one particular model being superior in performance to another. For example, both cropped and baseline images assessed by the customised VGG16 model achieved the same AUC result. However, the original VGG16 model utilised by Ferreira et al. [50] achieved an AUC of 85% with baseline images and 88% AUC with cropped images. Future studies should consider comparing cropped and uncropped images directly to make a definitive verdict. It would be unwise to conclude that AI models that are trained with the cropped ROI prior to testing with the dataset learn relevant feature representations toward classifying the task of interest without considering the probability of overfitting, reducing the generalisability of the results. In addition, cropping images before training involves more computational power and manpower.

When compared to the adapted Inception V3 architecture model employed by Kermany et al. [43], all four of the aforementioned outcome measures achieved greater results than the said platform in differentiating bacterial pneumonia from viral pneumonia on paediatric chest X-rays.

All five studies selected for this systematic review utilised the dataset obtained from Guangzhou Women and Children’s Medical Centre, China. The selection of said dataset by all five groups of authors immediately excluded participation bias and allowed each AI model to be accurately compared. The included radiographs by Rajaraman et al. also included ‘noisy images’ to reduce bias and improve the model generalisation [5]. All authors also developed AI models with multi-level architectures, which, unlike the study by Kermany et al., avoided limited prediction accuracy. However, using the same dataset from Guangzhou place the studies’ generalisability, as well as the algorithms’ abilities in detecting pneumonia using noisy images since only one study included noisy images, into question. On the other hand, the upper age limit in Gu et al.’s study was 9.7 years old, diminishing its comparability to other studies included in this review.

At present, there is no sufficient guidance in critically appraising machine learning prediction models. Thus, one of the limitations of this study is that the information in this review may not be able to be combined or pooled together with other systematic reviews on the same subject matter and compare the data and information directly.

In order to improve and generalise results achieved by AI models in the classification of pneumonia on paediatric CXRs, further datasets should be developed worldwide. This is because pneumonia caused by different bacteria/viruses has different radiographical appearances [53], while the prevalence of different risk factors and types of paediatric pneumonia varies in different countries [54,55]. An algorithm that can effectively detect/rule out pneumonia and its ability to identify pneumonia aetiology can be the single best tool to be employed in the effort of reducing global paediatric pneumonia mortality. This is particularly crucial for resource-restrained countries with limited radiographic reporting capacity. The said datasets should include a variety of normal paediatric CXRs, as well as CXRs that belonged to those who had clinically confirmed having bacterial or viral pneumonia. Diagnosis should be confirmed by seasoned radiologists and other diagnostic test results, such as sputum cultures. This would improve the generalisation of AI models used to classify aetiologies. Finally, a major barrier impeding the translation of these results to a clinical setting is the comparison of said results to reporting clinicians’ reports. Each of the included articles in this study assessed AI algorithms against one another rather than comparing outcome measures to that of consultant radiologists, and this would be a great research question for future investigations. One study assessed the performance of an AI model to that of a number of radiologists in the classification of viral versus bacterial pneumonia in paediatric chest X-rays [56]. However, this study only assessed the differentiation of the two pneumonic aetiologies and not normal vs. pneumonic, so none of these algorithms could replace human radiologic interpretation currently [56]. In order for AI to translate into the clinical setting, future studies should compare different AI algorithms in differentiating normal versus bacterial versus viral pneumonia from that of human radiologists. AI algorithms are normally trained on one specific modality and on a specific pathology, while human radiologists have a basic and fundamental knowledge of all modalities and common pathologies then often specialise in one, sometimes multiple, organ systems.

There are only a few commercially available AI product that is capable of interpreting both chest and appendicular musculoskeletal X-ray images at the time of writing. This algorithm can detect seven different pathologies (fracture, pleural effusion, lung opacification, joint effusion, lung nodules, pneumothorax, and joint dislocation). A recent study which compared this AI algorithm’s ability to human radiologists showed that the algorithm failed to pass the Fellowship of Royal College of Radiologists Rapid Reporting examinations [57]. This exam is normally taken between radiology speciality training years four and five in the UK [58].

This review has limitations. The age range used in these studies varied, making direct comparisons between the AI algorithms difficult. Secondly, only five studies are included in this review. Although all five studies used the same dataset from Guangzhou, the fact that some studies included noisy images and others included cropped images made it impossible to compare the algorithms on par with each other, let alone with other algorithms that are trained with a different dataset. Finally, the number of images that these algorithms were trained and validated on were small, compared to some of the commercially available algorithms [59,60,61,62,63,64,65] and some algorithms that are at the research stage [66].

Since the accuracy for all algorithms is so high, the amount of data is limited. Concerns about overfitting arise, which commonly occur due to the small training dataset [67,68]. Overfitting is a phenomenon where the algorithm fits in all the noises within the dataset, and the algorithm memorises all the peculiarities, finding the pattern best fit to the training data but not the general prediction trend, which is the goal of training [69]. Overfitted AI models are only applicable to the training dataset, but not the unseen dataset, losing the generalisability of the prediction [70,71,72]. Going forward, these algorithms should be further scrutinised. If overfitting did occur during training, the problem would need to be addressed in further studies before using these algorithms in clinical practice. Overfitting is an innate problem in using AI in radiology. First, there is no set threshold for a “sufficient” amount of data [70]. The more images and data fed into the algorithm for training, the less likely for overfitting to happen [49,73,74]. However, a large number of images can be very difficult to obtain in medicine due to confidentiality issues. Even if these data protection hurdles are overcome, the financial cost of acquiring medical images or data can be substantial [75,76]. Adding sub-optimal images into the training set, such as rotated or images with artefacts, can minimise the chance of overfitting [77,78] while increasing the overall image counts in the training set [78,79] can also mimic the situations in clinical practice.

This review gave the radiology community an insight into how AI can help us to reduce paediatric pneumothorax mortality rate and pose a potential solution to replace or to add on top of the current teleradiology system, especially in low- to middle-income or rural areas. This review took the essential step in starting the conversation in considering how radiology can utilise AI to improve workflow, but indeed, this is a start of a very long journey of research before AI can be used clinically. In this context, AI has the potential to assist in the classification of the aetiologies of pneumonia and, therefore, greatly increase treatment rates, potentially saving lives. The future landscape and the scope of practice of the radiology workforce are both going towards an exciting trajectory. “In the Age of the Algorithm, humans have never been more important” [80].

## 5. Conclusions

A number of the AI models achieved high accuracy in differentiating paediatric pneumonia. This showed potential in the automated classification of paediatric pneumonia on CXRs. Future studies should involve the comparison of AI models to that of a radiologist. Future research should focus on advancing the use of AI in identifying paediatric pneumonia in a clinical environment by including more sub-optimal images to ensure AI can correctly and accurately identify paediatric pneumonia in different circumstances and situations.

## Figures and Tables

**Figure 1 children-10-00576-f001:**
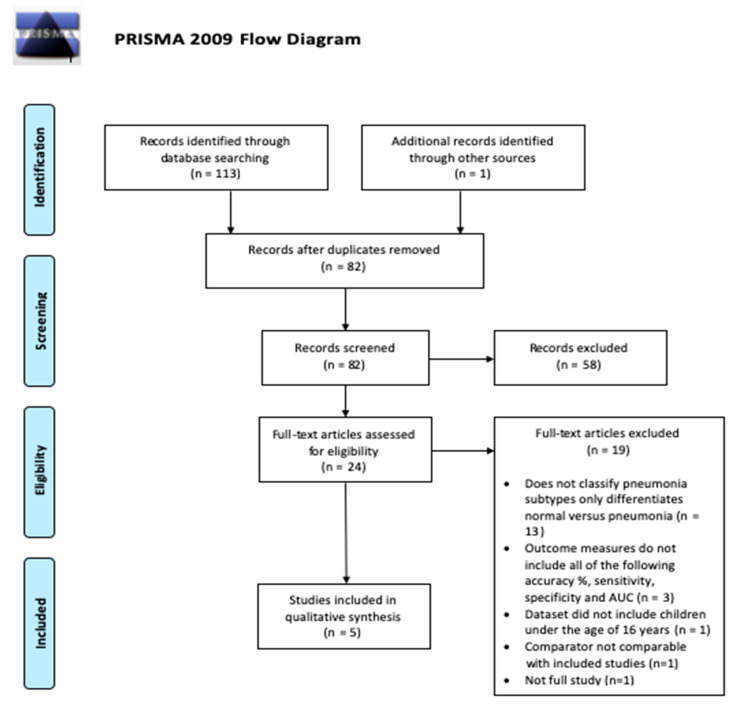
A PRISMA flow chart illustrating the filtering and gathering of eligible studies.

**Figure 2 children-10-00576-f002:**
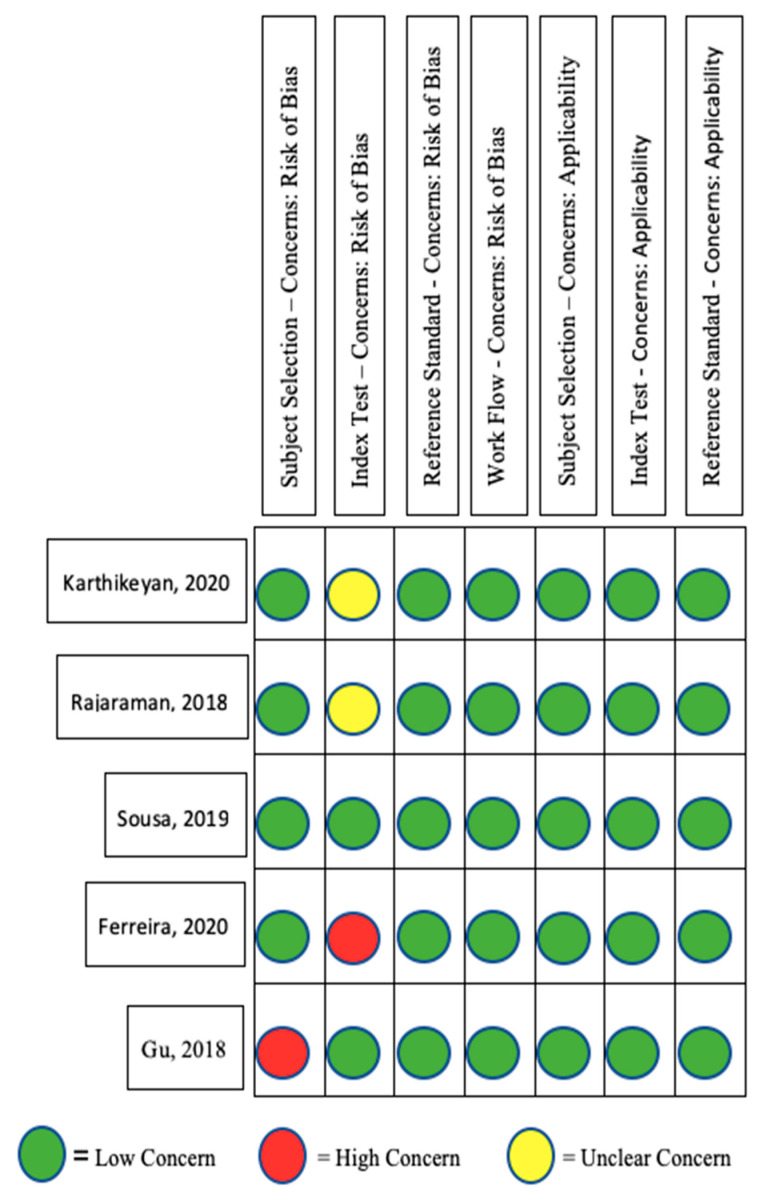
A risk of bias table of the five included studies from the adapted version of the QUADAS-2 ROB tool with a three-point scale indicating low, high and unclear concern [5,44,49,50,51].

**Table 1 children-10-00576-t001:** Characteristics of the AI algorithm employed, pre-processing methods utilised, as well as detailed descriptions of each dataset used in each of the included studies. Abbreviations: N/A (not applicable), ROI (region of interest), CLAHE (contrast limited adaptive histogram equalisation), CNN (convolutional neural network).

*Author, Year*	*Algorithm Employed*	*Type of AI*	*Age of Participants* *(Years)*	*No. of Images (by Aetiology)*					*No. of Images Used in*			*Pre-Processing Methods*
				*Normal*	*Pneumonia*			Total	Training	Validation	Testing	
					*Bacterial*	*Viral*	*Total*					
*Gu et al., 2018 [44]*	AlexNet, 3 Handcrafted Features (Gray level co-concurrence matrix-based feature extraction, Haar wavelet transform feature extraction, Histogram of oriented gradient-based feature extraction)	DL	5.5 ± 4.2	0	2665	1848	4513	4513	3211	802	500	All input ROIs images were resized to 256 × 256 matrices. An AlexNet-based fully convolutional networks model was applied for the segmentation of lung regions.
*Ferreira et al., 2020 [50]*	VGG16, Inception V3 architecture	DL	1–5	1349	2538	1345	3883	5232	5232	0	624	CLAHE method Chest cavity cropped from radiograph images Combination of the above
*Sousa et al., 2019 [51]*	CNN, Inception V3 Architecture	DL	1–5	1349	2538	1345	3883	5232	624	0	624	All of the images were re-dimensioned to a 300 × 300 pixels resolution and saved in a one-dimensional format
*Rajaraman et al., 2018 [5]*	Sequential CNN, Residual CNN, Inception CNN, Customised VGG16	DL	1–5	1349	2538	1345	3883	5232	5232	0	624	Both baseline and cropped images are resampled to 1024 × 1024 pixel Cropped using an algorithm based on anatomical atlases to detect the lung ROI automatically
*Karthikeyan 2020 [49]*	AlexNet, ResNet18, DenseNet201, SqueezeNet	DL	1–5	1341	2561	1345	3906	5247	4500	0	398	The images were resized to 227 × 227 pixels

**Table 2 children-10-00576-t002:** The diagnostic accuracy of each AI algorithm employed in its respective paper in terms of sensitivity, specificity, accuracy, and AUC (area under the curve). The highest value for each outcome measure is highlighted in bold. ± indicates the standard deviations.

Author, Year	Algorithm	Sensitivity	Specificity	Accuracy	AUC
*Gu et al.,* *2018 [44]*	AlexNet (DCNN ONLY)	0.6322 ± 0.0023	0.7072 ± 0.0023	0.7360 ± 0.0023	0.7384 ± 0.0023
	GLCM Features	0.6378 ± 0.0058	0.8980 ± 0.0062	0.7060 ± 0.0672	0.7060
	Wavelet Features	0.5612 ± 0.0065	0.8779 ± 0.0205	0.6769 ± 0.0100	0.6769
	HOG Features	0.5714 ± 0.0617	0.8651 ± 0.0664	0.7511 ± 0.0127	0.6930
	All Handcrafted Features	0.6213 ± 0.0482	0.8848 ± 0.0387	0.7640 ± 0.0330	0.7200 ± 0.0060
	Fused Features (DCNN + all handcrafted features)	0.5567 ± 0.0379	0.9267 ± 0.0301	0.7692 ± 0.0122	0.8234 ± 0.0014
*Ferreira et al., 2020 [50]*	VGG16 and Baseline Set	*Not Stated*	*Not Stated*	*Not Stated*	0.85
	VGG16 and Set A	*Not Stated*	*Not Stated*	*Not Stated*	0.88
	VGG16 and Set B	*Not Stated*	*Not Stated*	*Not Stated*	0.83
	VGG16 and Set C (ensemble set)	**0.963**	0.851	0.921	0.91
	Inception V3 architecture	0.886	0.909	0.907	0.940
*Sousa et al.,* *2019 [51]*	‘Best generated model’	0.913	0.696	0.831	0.831
	Inception V3 architecture	0.886	0.909	0.907	0.940
*Rajaraman et al., 2018 [5]*	Sequential CNN—Baseline	*Not specified*	0.838	0.928	0.954
	Residual CNN—Baseline	*Not specified*	0.784	0.897	0.921
	Inception CNN—Baseline	*Not specified*	0.714	0.854	0.901
	Customised VGG16—Baseline	*Not specified*	0.860	0.936	**0.962**
	Sequential CNN—Cropped	*Not specified*	0.838	0.928	0.956
	Residual CNN—Cropped	*Not specified*	0.798	0.908	0.933
	Inception CNN—Cropped	*Not specified*	0.730	0.872	0.919
	Customised VGG16—Cropped	*Not specified*	0.860	0.936	**0.962**
*Karthikeyan,* *2020 [49]*	AlexNet	0.94	0.845	0.90	0.89
	ResNet18	0.92	0.82	0.87	0.87
	DenseNet201	0.96	**0.94**	**0.95**	0.952
	SqueezeNet	0.905	0.75	0.83	0.83

## Data Availability

Please email the first author.

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
