# Peer review of "Efficacy of Artificial Intelligence in the Categorisation of Paediatric Pneumonia on Chest Radiographs: A Systematic Review"

_children, 2023, doi:10.3390/children10030576_

Round 1

Reviewer 1 Report

The aim of this manuscript focuses on the classification of pediatric pneumonia according to a chest x-ray. However, I didn't find the categorization of pneumonia based on the chest x-ray using an AI in this manuscript. The authors may define the categorization of pneumonia in the method section and explain the finding in the result.

The other way is changing the title and the aim, then correlating it with the finding.

Author Response

please see the revision

Reviewer 2 Report

Authors are stick with classification accuracy (specially in conclusion section) please add other important matrixes (recall, precision, f1-score etc.) and add your own conclusions few more lines in conclusion section.

Author Response

please see the revision

Reviewer 3 Report

In this paper, authors have done some literatures review on the application of AI on classification of paediatric pneumonia on Chest radiographs. While the writing of this paper is good, I believe the contribution of this paper is not enough and there is lack of novelty in this paper. 

Author Response

please see the revision 

Reviewer 4 Report

• The abstract should mention significance of your study, like why this topic is important, method used why etc.

• The introduction is not clear and very less literature is used. Follow this instruction: The introduction should briefly place the study in a broad context and highlight why it is important. It should define the purpose of the work and its significance, including specific hypotheses being tested. The current state of the research field should be reviewed carefully, and key publications cited. Please highlight controversial and diverging hypotheses when necessary. Finally, briefly mention the main aim of the work and highlight the main conclusions. Keep the introduction comprehensible to scientists working outside the topic of the paper.

• Below papers has some interesting implications that you could discuss in your introduction and how it relates to your work.

• Praveen, S.P.,. et al. ResNet-32 and FastAI for diagnoses of ductal carcinoma from 2D tissue slides. Sci Rep 12, 20804 (2022). https://doi.org/10.1038/s41598-022-25089-2

• Vulli, A.; et al.. Fine-Tuned DenseNet-169 for Breast Cancer Metastasis Prediction Using FastAI and 1-Cycle Policy. Sensors 2022, 22, 2988

• In the introduction, what key theoretical perspectives and empirical findings in the main literature have already informed the problem formulation? What major, unaddressed puzzle, controversy, or paradox does this research address? 

• It would be interesting if the authors report the trade-off compared to other methods especially the computational complexity of the models. Some techniques require more memory space and take longer time, please elaborate on that. 

• Authors should further clarify and elaborate novelty in their contribution.

• What are the limitations of the present work?

Author Response

please see the revision

Round 2

Reviewer 1 Report

None

Reviewer 3 Report

The manuscript still needs revision to improve the quality such as the style, writing and the formation. 

Reviewer 4 Report

.